# Laser harmonic generation with independent control of frequency and orbital angular momentum

Raoul Trines [1] ✉, Holger Schmitz [1], Martin King [2,3], Paul McKenna [2,3] & Robert Bingham [1,2]

The non-linear optical process of laser harmonic generation (HG) enables the creation of high quality pulses of UV or even X-ray radiation, which have many potential uses at the frontiers of experimental science, ranging from lensless microscopy to ultrafast metrology and chiral science. Although many of the promising applications are enabled by generating harmonic modes with orbital angular momentum (OAM), independent control of the harmonic frequency and OAM level remains elusive. Here we show, through a theoretical approach, validated with 3D simulations, how unique 2-D harmonic progressions can be obtained, with both frequency and OAM level tuned independently, from tailored structured targets in both reflective and transmissive configurations. Through preferential selection of a subset of harmonic modes with a specific OAM value, a controlled frequency comb of circularly polarised harmonics can be produced. Our approach to describe HG, which simplifies both the theoretical predictions and the analysis of the harmonic spectrum, is directly applicable across the full range of HG mechanisms and can be readily applied to investigations of OAM harmonics in other processes, such as OAM cascades in Raman amplification, or the analysis of harmonic progressions in nonlinear optics.

The generation of harmonic radiation (HG) with higher-order phase or polarisation topology, especially Laguerre-Gaussian (LG) modes with orbital angular momentum (OAM), opens up a vast panorama of novel applications such as quantum computing[1], enhanced optical communications[2], super-resolution microscopy[3] and optical tweezers[4]. The underpinning physics enables the detection of light-beam OAM spectra to be used to diagnose the existence of rotating massive astrophysical objects such as black holes[5].

When a single laser beam is used to generate the harmonics, the OAM level of a given harmonic frequency is often linked to its harmonic order. This involves both the transfer of OAM from a pump laser beam to harmonics in gas targets[6–11] and HG using a circularly polarised (CP) beam without OAM, through exploiting spin-to-orbital momentum conservation in laser-solid interactions[12–15] or laser-aperture

interactions[16–19]. In all of these configurations, the harmonic generation follows the following principles: (i) no energy, linear or angular momentum can be left behind in the medium or target because the laser interaction with the medium is isotropic[20,21]; (ii) HG in an isotropic medium (gas) conserves spin and orbital angular momenta separately within the EM waves, so HG by pure CP waves is not possible as it would violate spin conservation, while the OAM of harmonics generated by linearly polarised (LP) beams carrying OAM will be proportional to the harmonic frequency[14]; (iii) HG in laser-solid interactions conserves total angular momentum within the EM waves, and so a single CP wave without OAM can produce CP harmonics with OAM via spin-to-orbital angular momentum conversion[12,13,16].

In this article, we demonstrate that when a CP laser pulse interacts with a target with a defined periodic surface structure, angular

[1]Central Laser Facility, STFC Rutherford Appleton Laboratory, Didcot OX11 0QX, United Kingdom. [2]Department of Physics, SUPA, University of Strathclyde, Glasgow G4 0NG, United Kingdom. [3]The Cockcroft Institute, Sci-Tech Daresbury, Warrington WA4 4AD, United Kingdom. ✉e-mail: raoul.trines@stfc.ac.uk

momentum can be left behind in the target, as in a q-plate[22,23]. The OAM content of the harmonics can then differ significantly from the prediction of the simple spin angular momentum (SAM) to OAM conversion, and we exploit this behaviour to independently control the frequency and OAM level of the harmonics. In parallel to this, we introduce a systematic approach for the description of simultaneous HG in multiple variables (e.g., frequency $\omega$ and OAM level $\ell$) where all fields are decomposed into modes with pure CP, constant amplitude and definite spin $\sigma = \pm 1$, with the harmonic spectra given as functions of "signed" frequencies and wave vectors $\omega/\sigma$, $\mathbf{k}/\sigma$ instead of $\omega$, $\mathbf{k}$. We show that the harmonic progression in ($\omega/\sigma$, $\mathbf{k}/\sigma$) space is always a regular grid, defined by the ($\omega/\sigma$, $\mathbf{k}/\sigma$) spectrum of the initial pump-target configuration, and tuneable via both the pump laser composition and the target structure: two initial modes generate regularly spaced points on a single line and three independent modes drive a regular 2-D grid. This important finding generalises previous results on HG involving a "bicircular" pump configuration[10,24–26] (where the spectral peaks are forced onto a single line, but not regularly spaced), and can also explain findings on CP pumps (2 initial peaks, 1-D spectral

grid) versus LP pumps (4 initial peaks, 2-D spectral grid) reported in Hickstein et al.[27]. To verify the above predictions of regular spectra, we have developed dedicated diagnostic methods to determine the ($\omega/\sigma$, $\mathbf{k}/\sigma$) spectrum from the results of 3-D numerical simulations of structured reflecting and transmissive targets. This approach demonstrates that a predictable, regular 1-D or 2-D harmonic progression can be generated, in ($\omega/\sigma$, $\ell/\sigma$) space, where both frequency $\omega/\sigma$ and OAM level $\ell/\sigma$ of the harmonics can be tuned independently. We also show how the 2-D harmonic progressions with sufficiently large separation can be used to generate a tuneable frequency comb in which all frequencies have circular polarisation. Our approach unifies fields as diverse as HG in gases, crystals and solid targets[10,11,24,27], and even Raman scattering[28] and laser beat waves[29,30].

## Results

The essence of our HG scheme and corresponding data analysis is shown schematically in Fig. 1. A relativistically intense CP pump pulse is incident on a target with either a periodically structured indentation (Fig. 1a) or aperture (Fig. 1b). Shaped targets with a periodicity of three

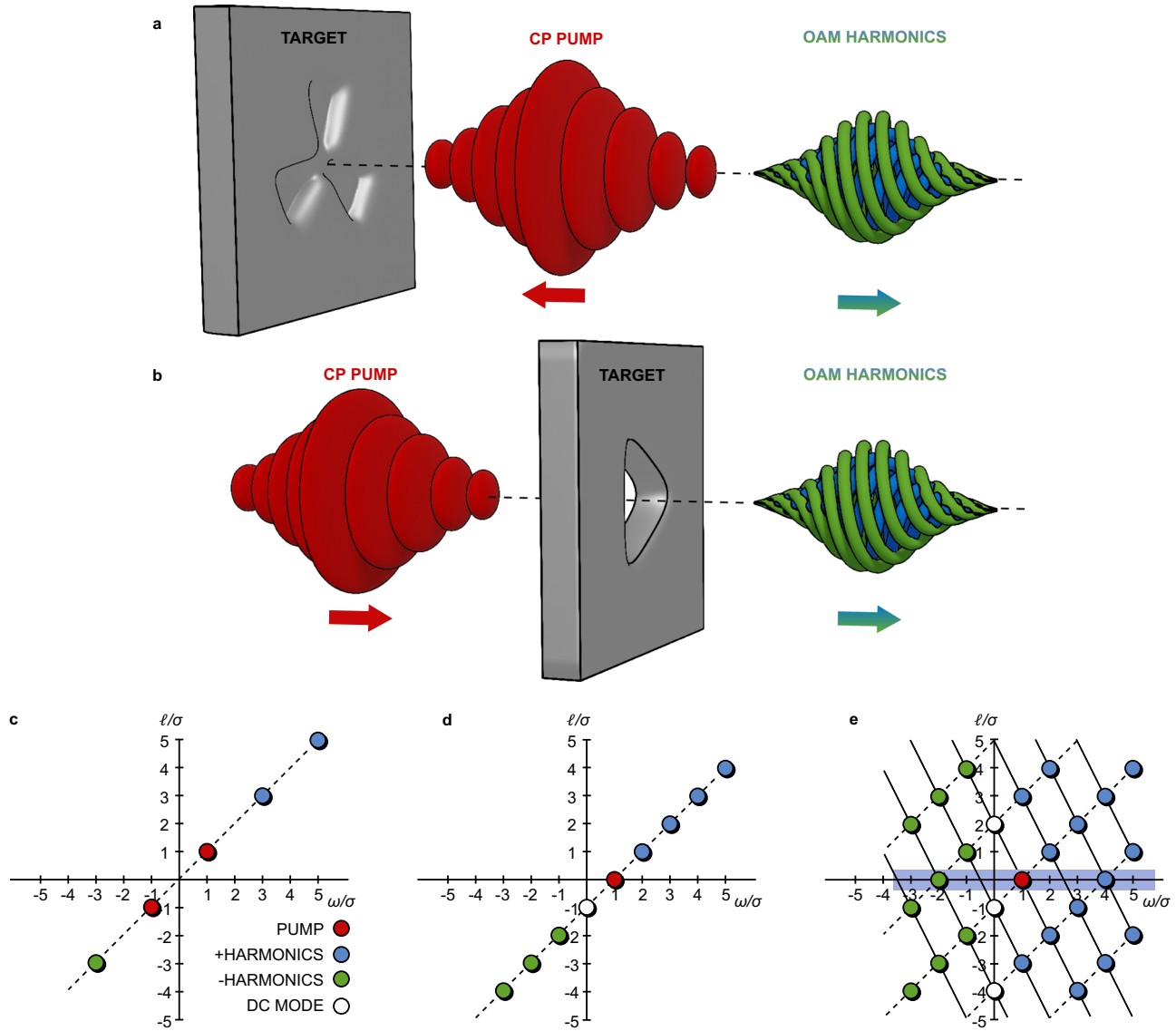

**Fig. 1 | Schematic illustrating the generation of higher-order harmonics.** Shown are the LCP ($\sigma = 1$, blue) and RCP ($\sigma = -1$, green) harmonics of an incoming CP pump pulse (red) in **a** reflection and **b** transmissive configurations, from a threefold structured target. **c**–**e** Expected 2D harmonic spectrum from: **c** a flat target driven by an LP pump pulse with $\ell = 1$; **d** a circular aperture or dent target driven by a CP pump pulse of $\ell = 0$; and, **e** a threefold reflective or transmissive target with a CP pump pulse of $\ell = 0$. The blue shaded area indicates the potential for a frequency comb at $\ell = 0$.

are shown in this example. The irradiated target surface oscillates relativistically, generating harmonics that are reflected or transmitted.

To visualise the resultant 2-D harmonic progression, for a real-valued CP field $(E_x, E_y) = (\cos(\omega t/\sigma), \sin(\omega t/\sigma))$, we find that $E_x + iE_y = \exp i(\omega t/\sigma)$ (see Supplemental section S1 for more detail). Our data analysis approach involves calculating $\Psi(t, r, \varphi) = E_x + iE_y$, integrating over $r$ and performing a 2-D signed Fourier transform to obtain the "signed" spectrum $\tilde{\Psi}(\omega/\sigma, \ell/\sigma)$. The three main advantages of this approach are: (i) harmonic peaks are on a 1-D line or in a 2-D regular grid; (ii) CP harmonics with helicity opposite to that of the pump are not masked by the CP harmonics with the same helicity as the pump; (iii) many aspects of the harmonic progression, like the regular steps and the dimensionality of the progression, symmetries and conserved quantities can be verified immediately from the graphical representation of the data.

Figure 1c−e shows how various HG schemes can be represented in a 2-D CP $(\omega/\sigma, \ell/\sigma)$ spectrum. Figure 1c shows typical HG with an LP pump laser pulse, incident on a planar target, with OAM level $\ell = 1$; the pump is decomposed into two symmetric CP modes represented by the points (1, 1) and (−1, −1). The harmonic progression is then given by the sequence of equidistant points $\pm (2n + 1)(1, 1)$, which combine to yield all the odd LP harmonics with OAM level equal to the harmonic number[31]. Figure 1d represents the HG when CP laser light interacts with a circular dent[12,13] or a circular aperture[16,32]. The pump laser is represented by the point (1, 0) and the DC mode corresponding to the target shape by (0, −1). The harmonic spectrum is again given by a sequence of equidistant points $(n, n-1)$, which includes CP harmonics with opposite spin to the pump. These have been predicted, but never demonstrated[13]. Finally, Fig. 1e shows the spectrum of a CP pump laser interacting with a target with a three-fold surface structure (as in Fig. 1a, b). The target is represented by the points (0, −1) and (0, 2), and together with the pump laser at (1, 0), these points drive a 2-D regular grid of harmonic modes. The harmonic light on-axis, which preferentially selects the modes with $\ell = 0$, is a frequency comb of pure CP modes $\omega_n/\sigma_n = (2n + 1)\omega_0$. The spacing in the comb is fully tuneable by tailoring the periodicity of the structured target.

## Description and analysis of the harmonic progression

To describe this behaviour for laser-driven HG in an isotropic medium, the leading nonlinear term driving the process is usually cubic, e.g., $\propto A^2 \mathbf{A}$, where $\mathbf{A}$ is the vector potential of the laser field (see Supplemental Section S2 for conventions used throughout, and Supplemental Section S3 for a full description of harmonic progressions generated by a cubic nonlinear term). However, for a laser beam with oblique incidence (angle $\alpha$) onto a flat solid surface, the nonlinear term takes the form $|\mathbf{A} - \tan(\alpha)\mathbf{e}_x|^2(\mathbf{A} - \tan(\alpha)\mathbf{e}_x)$[33]. More generally, a solid target with a shaped surface can result in a nonlinear term $|\mathbf{A} - \mathbf{A}_{DC}|^2(\mathbf{A} - \mathbf{A}_{DC})$, where $\mathbf{A}_{DC}$ is a time-independent, but space-dependent vector field that will influence the topology of the generated harmonics (see section II B). The harmonics generated in Lichters et al.[33] (constant $\mathbf{A}_{DC}$) have a different topology than those generated in Wang et al.[12] or Li et al.[13] for a circular dent target ($\mathbf{A}_{DC} \propto \mathbf{e}_r$ in polar coordinates). A similar mechanism emerges when considering targets with a circular aperture, for which the role of $\mathbf{A}_{DC}$ is played by the normal vector of the inner aperture surface[16,17,19,34,35]. Here, we consider a variety of both reflective and aperture targets with shaped vector fields $\mathbf{A}_{DC}$ to achieve full control of the topology of the generated harmonic spectrum. Our approach has the following features:

(i) All EM fields are decomposed into pure CP modes, since these have constant $A^2$ and well-defined spin $\sigma$, in addition to frequency $\omega$ and wave vector $\mathbf{k}$.

(ii) We use signed frequencies and wave vectors $\omega/\sigma$, $\mathbf{k}/\sigma$ rather than $\omega$, $\mathbf{k}$, since $(\cos(\omega t), (1/\sigma)\sin(\omega t)) = (\cos(\omega t/\sigma), \sin(\omega t/\sigma))$ for a pure CP mode with $\sigma = \pm 1$, while two CP modes A and B beat as

$\mathbf{A}_A \cdot \mathbf{A}_B \propto \cos[(\omega_B/\sigma_B - \omega_A/\sigma_A)t]$. We use $\omega/\sigma$ rather than $\sigma\omega$ since $\omega/\sigma = \hbar\omega/(\hbar\sigma) = \mathcal{E}/S_z$ for a CP photon with energy $\mathcal{E}$ and spin $S_z$.

(iii) For a laser beam consisting of two CP modes A and B, the harmonics will be generated by the factor $2\mathbf{A}_A \cdot \mathbf{A}_B$, since $A_{A,B}^2$ are constant; therefore, the harmonic progressions for $\omega$ and $\mathbf{k}$ become ($n \in \mathbb{Z}$):

$$\omega_n/\sigma_n = \omega_A/\sigma_A + n(\omega_B/\sigma_B - \omega_A/\sigma_A) \quad (1)$$

$$\mathbf{k}_n/\sigma_n = \mathbf{k}_A/\sigma_A + n(\mathbf{k}_B/\sigma_B - \mathbf{k}_A/\sigma_A) \quad (2)$$

(iv) Eq. (1) can readily be extended to multiple dimensions. Two CP modes generate equidistant harmonics on a line, whereas three non-degenerate CP modes produce a regular 2-D grid, and four non-degenerate modes generate a regular 3-D lattice. Since all these grids are regular, this approach can be generalised to many different scenarios. In addition, this approach enables 1-D and 2-D harmonic progressions to be easily distinguished, which was not possible in previous works (e.g., ref. 10,27,36).

(v) The selection rules derived using either the "photon counting" approach[6,11,27,31,37–44] or the "symmetry-based" approach[45–52] can be derived from Eq. (1), thus providing a link between these two approaches.

(vi) The collection of symmetries governing the harmonic progression can be described using this approach, and a connection can even be made to the conserved quantities of Noether's Theorem, thus providing a generalised description of quantities like the "torus-knot angular momentum"[10,24–26].

(vii) In particular, we investigate targets that have $n$-fold rotational symmetry; their DC modes will be represented by $\ell_{DC}/\sigma_{DC} = -1$ and $\ell_{DC}/\sigma_{DC} = -q$, while $n = |1 - q|$ (see the Methods section for details).

Our approach assumes that harmonic modes can be fully separated by frequency, wave vector and spin. In practice, separation by frequency and direction of propagation are common, while separation by OAM level is possible but uncommon[53]. Separation by spin, i.e., left and right CP modes, is not practical, especially for higher frequencies, and not normally done in experiments. The consequences for our work are that whenever we identify two harmonic modes that have the same values of $\omega$ and $\ell$ and only differ by $\sigma$, the superposition of these modes yields a mode with elliptic polarisation (EP) at $\omega$ and $\ell$. This may happen when we use targets with low rotational symmetry and the peaks in the harmonic spectrum are too closely spaced; see the discussion of our simulation results in Section II C for more details.

## The DC Mode in laser-solid interactions

To describe HG in laser-solid interactions, such as those considered in the main text, we use the model by Lichters et al.[33], which details the interaction of a laser beam with a solid surface. For oblique incidence at an angle $\alpha$ with respect to the target normal, where the projection of $\mathbf{k}$ on the target surface is given by $k \sin(\alpha)\mathbf{e}_x$, the model equations are as follows ($\mathbf{a}_{DC} = \tan(\alpha)\mathbf{e}_x$):

$$(\partial_t^2 - c^2\nabla^2)(\mathbf{a} - \mathbf{a}_{DC}) = -\omega_p^2 \frac{n}{\gamma \cos\alpha}(\mathbf{a} - \mathbf{a}_{DC}), \quad (3)$$

$$n/\gamma \approx 1 + \delta n - (\mathbf{a} - \mathbf{a}_{DC})^2/2, \quad (4)$$

$$\partial_t^2 \delta n = -\frac{c^2}{2\gamma^2}\nabla^2(\mathbf{a} - \mathbf{a}_{DC})^2, \quad (5)$$

$$\gamma^2 = 1 + (\mathbf{a} - \mathbf{a}_{DC})^2. \quad (6)$$

Similar to above, we define $\Psi = (a_x - \tan\alpha) + ia_y$ and only retain leading-order terms for small wave amplitudes. Then, for pump laser with frequency $\omega_0$ and wave number $k_0$ we can rewrite the above system as:

$$\cos\alpha\left(\partial_t^2 - c^2\nabla^2\right)\Psi + \omega_p^2\Psi = -\frac{1}{2}\omega_p^2\left(1 + c^2k_0^2/\omega_0^2\right)(\Psi^*\Psi)\Psi, \quad (7)$$

which has the required shape with a cubic nonlinear term. However, this requires that $\Psi$ does not just cover the EM vector potential $\mathbf{a}$, but also a synthetic "DC mode" $\mathbf{a}_{DC} = \tan\alpha\,\mathbf{e}_x$, which is a consequence of the laser-target geometry. The DC mode has $\omega = 0$, so its spin $\sigma$ is undefined, but $\mathbf{k}/\sigma$ and $\ell/\sigma$ are well-defined via the mode's spatial dependence. The presence of this DC mode explains why a single laser beam with circular polarisation can generate harmonics when hitting a target at an oblique angle[33]: unlike in HG in gas, the circularly polarised mode is not on its own but can beat against the DC mode.

The DC mode for oblique laser incidence onto a planar target has $\Psi_{DC} = \tan\alpha = \mathrm{const}$, which means that $\mathbf{k}/\sigma = \ell/\sigma = 0$. It should also be regarded as having linear polarisation in $x$. With this, judicious application of Eq. (1) will yield all the selection rules described in ref. 33. Similarly, the DC mode for a cone-shaped dent[12,13] takes the form $\mathbf{A}_{DC} \propto \mathbf{e}_r$ or $\Psi_{DC} \propto \exp(i\varphi)$, corresponding to $\omega/\sigma = 0$, $\ell/\sigma = -1$, and the selection rules from these works can all be recovered from Eq. (1).

The same strategy with DC modes can be used to describe, e.g., second-harmonic generation in nonlinear materials, see, e.g., ref. 54. The cubic term in the equation for $\mathbf{a}$ can be rewritten as either $(\mathbf{a} - \mathbf{a}_1)^2(\mathbf{a} - \mathbf{a}_1)$ or $(\mathbf{a} - \mathbf{a}_1)^2(\mathbf{a} - \mathbf{a}_2)$ or $[(\mathbf{a}-\mathbf{a}_1)\cdot(\mathbf{a}-\mathbf{a}_2)](\mathbf{a}-\mathbf{a}_3)$. If one then equips the DC modes with the right higher-order structure (e.g., OAM), the second-harmonic light can then be generated with intrinsic OAM levels or other higher-order structure.

## Simulation results

Figure 2 shows examples of 3-D simulation results for harmonic generation by a CP pump laser passing through either a circular or a "trefoil" (threefold periodicity) aperture. Figure 2a–b shows the laser-target configuration and the OAM topology of the second-harmonic light generated during the interaction, as a function of time over one laser period. Figure 2c shows the 2-D ($\omega/\sigma$, $\ell/\sigma$) spectrum for the circular aperture case, with the second-harmonic modes ringed: a CP mode with the same (opposite) helicity to the pump and OAM level $\ell = 1$ ($\ell = 3$). Figure 2d shows the corresponding 2-D spectrum for the trefoil target; multiple CP modes are generated at $\omega/\sigma = 2\omega_0/\sigma_0$ ($\omega/\sigma = -2\omega_0/\sigma_0$) with the same (opposite) helicity to the pump and OAM levels $\ell = 3n + 1$ ($\ell = 3n$), $n \in \mathbb{Z}$.

Figure 2a–b shows the oscillatory Cartesian electric field components in space and time for the combined generated $2\omega_0$ harmonic direct from the simulations. As the analysis has been performed considering circular polarisation components, it is also useful to compare the Cartesian spatial distribution of the $-\omega$ and $+\omega$ components of the $2\omega_0$ harmonic ($\tilde{\Psi}(\omega = \pm 2\omega_0, x, y)$). This can be observed, respectively, in Fig. 3a–b for $n = 0$ and similarly in Fig. 3c–d for $n = 3$. The distributions for $n = 0$ show the clear annular structures expected for the predicted clean $l = 1$ and $l = 3$ modes in $+\omega$ and $-\omega$, respectively. These structures form at distinct radii, which may aid in detection. The $n = 3$ case is again more complex due to the interference of the different spatial orders. Of note is the on-axis central component in the $-\omega$ case that will have no OAM and will be discussed later. Up to this point, we have been considering the near-field profile of the pulse. By performing a 2-D spatial Fourier transform, it is possible to construct the far-field pattern in k-space ($\tilde{\Psi}(\omega = \pm 2\omega_0, k_x, k_y)$). This is shown in Fig. 3e–h for the same cases, respectively. For $n = 0$, there is no significant change in the spatial distribution of the 2nd harmonic light. For $n = 3$, the profile is significantly less complex with the distinct central feature on-axis for $-\omega$ and a triple lobed pattern in $+\omega$. It is expected that the 2D harmonic spectrum does not vary when considering the far-field and tests have been conducted using a moving window simulation technique to verify this, as shown in Supplemental section S4. Example results exploring the effects of varying the radius and corrugation depth of the target are presented in Supplemental section S5.

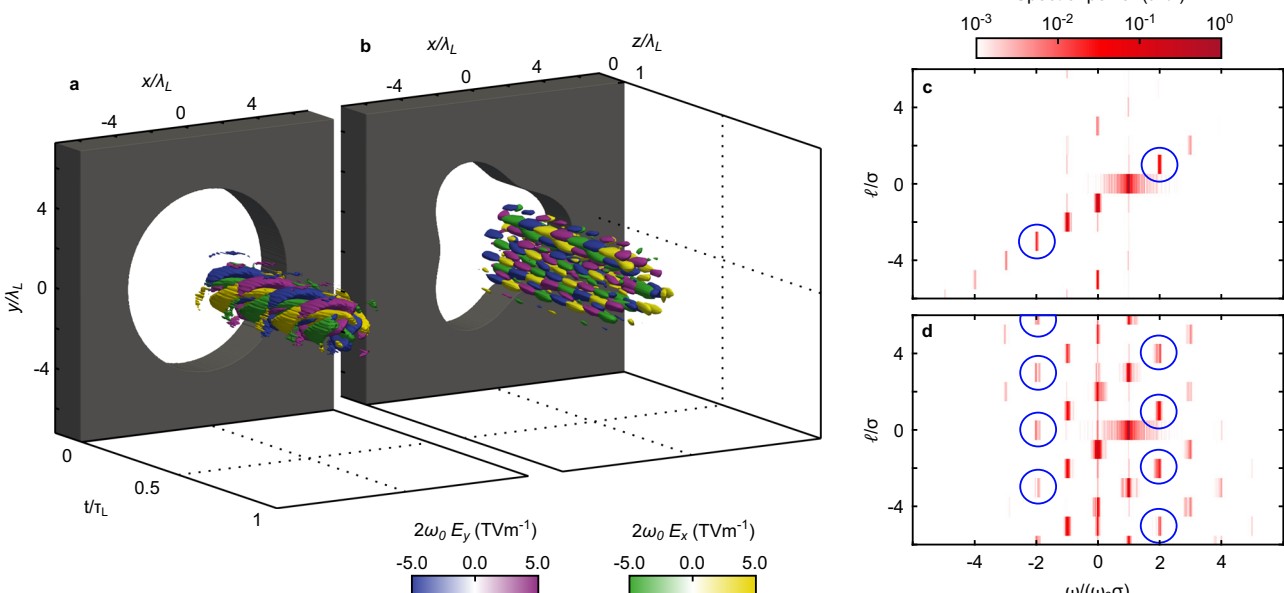

**Fig. 2 | Simulation results for second-harmonic generation in an aperture target.** The 3D simulation results showing the temporal behaviour (over 1 fundamental wave period $\tau_L$) of the generated orthogonal electric field components ($E_x$ and $E_y$) for $\omega = 2\omega_0$, from the interaction of a circularly polarised pump laser pulse of $\ell = 0$ with a target comprising of a: **a** circular (0-fold) aperture; and, **b**, structured (3-fold) aperture. These example results correspond to $x = 5\lambda_L$ and $t = 0$ indicates the time when the peak of the pump laser pulse reaches this position. The target profiles are indicated in grey. **c** 2D harmonic spectrum of the results for the target in **a**. **d** corresponding results for the target in **b**. The blue circles indicate the OAM levels present at $\omega = 2\omega_0$.

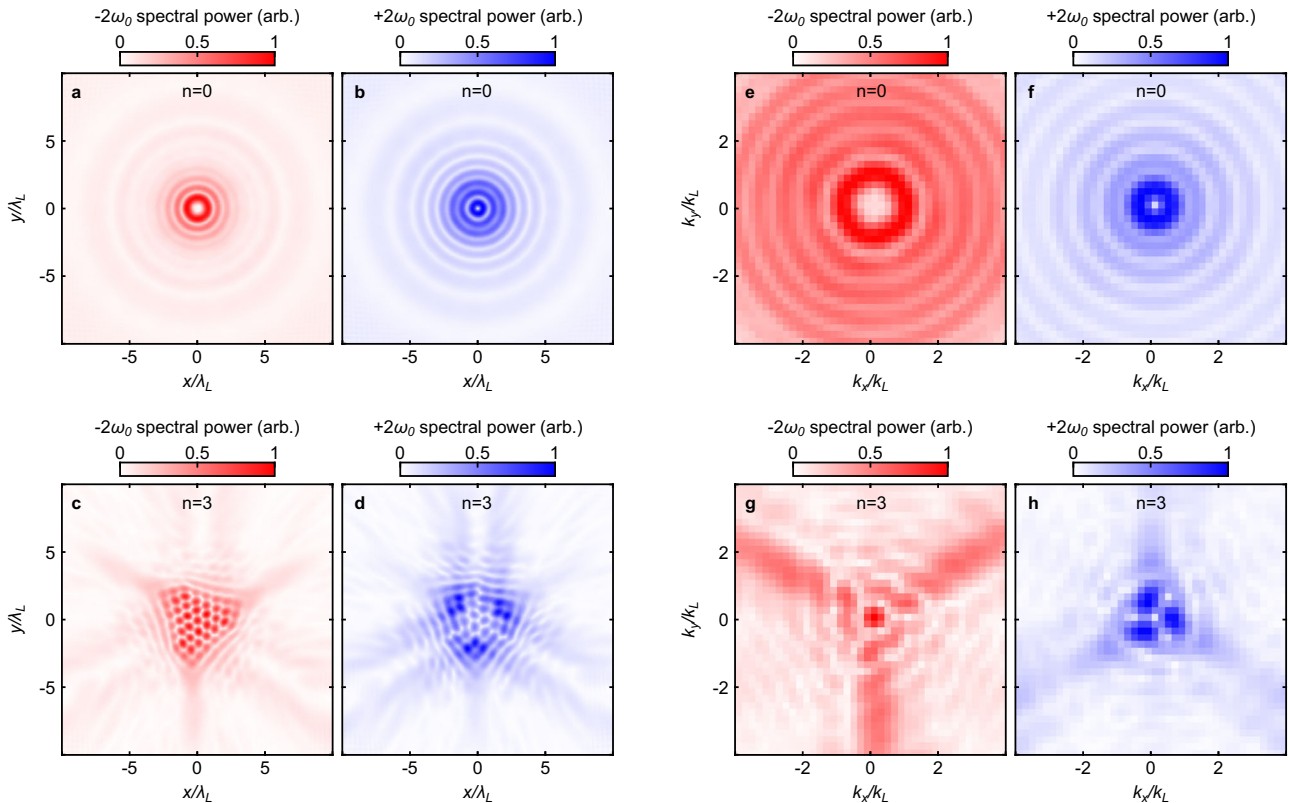

**Fig. 3 | Simulated spatial profiles for the second-harmonic frequency components. a–b** Spatial profile of the generated $-2\omega$ (**a**) and $+2\omega$ (**b**) light for the circular ($n = 0$) aperture. **c**, **d** Respective profiles for the structured ($n = 3$) aperture. **e–h** k-space profile for the same four cases (**e**: $-2\omega$, $n = 0$; **f**: $+2\omega$, $n = 0$; **g**: $-2\omega$, $n = 3$; **h**: $+2\omega$, $n = 3$). These profiles are sampled at $Z = 5\lambda_L$.

**Reflective target results.** We start with the analysis of the simulations involving "reflective" targets. Due to the fact that for the reflective targets the forward and backward waves need to be separated, not all predicted modes are observable. The specific height functions $h(x, y)$ of the targets used are shown in Fig. 4a–c for $q = 1$, $-1$ and $-4$, respectively. In Fig. 4d, the ($\omega/\sigma$, $\ell/\sigma$) spectra of the reflected light for a CP pump laser with spin $\sigma_0 = 1$ and no OAM ($\ell_0 = 0$), corresponding to the point $(1, 0)$, is shown. The $q = 1$ cone-shaped dent leads to a DC mode $\mathbf{A}_{DC} \propto \mathbf{e}_r$ or $\Psi_{DC} \propto \exp(i\varphi)$ with $\ell/\sigma = -1$, corresponding to the point $(\omega/\sigma, \ell/\sigma) = (0, -1)$. These two points are expected to define the harmonic progression $(\omega/\sigma, \ell/\sigma) = (n, n, -1)$, which is found in the 2-D Fourier spectrum from the simulation. Note that "negative" harmonics with spin opposite to that of the pump beam are observed; since their OAM level is different from the corresponding "positive" harmonics, they are genuine harmonics and not numerical echoes. While these have been predicted in high-power laser-solid interactions by Li et al.[13], they are demonstrated here for the first time. One can vary the values for $\sigma_0$ and $\ell_0$ so the pump corresponds to the point $(1/\sigma_0, \ell_0/\sigma_0)$, and this results in a harmonic progression $(n/\sigma_0, n(1 + \ell_0/\sigma_0) -1)$, defined by the points $(1/\sigma_0, \ell_0/\sigma_0)$ and $(0, -1)$. Due to the resolution and fourfold symmetry of the simulation grid, numerical artifacts are introduced as a result. These spectral peaks are found off the (dashed) diagonal line in Fig. 4d.

Figure 4e–f shows the resultant spectrum when using structured targets with $q = -1$ and $q = -4$, corresponding to the points $(0, +1)$ and $(0, +4)$, respectively. We observe that the point $(0, -1)$ is still present in both cases, along with several additional peaks. Together with the point $(1/\sigma_0, \ell_0/\sigma_0)$ from the pump, these new peaks form a 2-D grid of harmonic modes, as predicted by the model (see 2-D grid in Fig. 4e, f). The vertical spacing is controlled directly by the value of $q$ of the target. The shape of the 2-D grid of harmonic modes can be controlled

by $\sigma_0$ and $\ell_0$ of the pump mode. We note that modes with $\ell = 0$ occur for only specific values of $\omega/\sigma$. By selecting the on-axis harmonic light (where modes with $\ell = 0$ dominate), one can obtain a controlled "frequency comb"[11,45,49,55]; the vertical OAM spacing being converted into a horizontal frequency spacing (see also Section II D).

**Aperture target simulation results.** For the simulations involving aperture targets, Fig. 4g–i show the initial electron densities in the $(x, y)$-plane for apertures with no periodic variation or periods of $n = 2$ and $n = 5$ (equivalent to $q = -1$ and $q = -4$), respectively, at $z = 0.5\lambda_L$. The resultant ($\omega/\sigma$, $\ell/\sigma$) spectra of the generated light from aperture periods of $n = 0$, 2 and 5 ($q = 1$, $-1$, $-4$) are shown in Fig. 4j–l, respectively. Note that $\mathbf{n}$ for these apertures has the same symmetry as $\nabla h$ for the reflective targets displayed in Fig. 4a–f, which is observed in the spectra.

Again, note the presence of negative harmonics with spin opposite to that of the CP pump. These are clearly present in the harmonic light and can be observed via Fourier analysis of the function $\Psi = E_x + iE_y$. Especially the negative harmonics at $\omega/\sigma = -\omega_0/\sigma_0$, which would otherwise be masked by the transmitted light of the pump.

In Fig. 4e and k, a rotation by $180°$ around the origin will map spectral peaks onto other spectral peaks; this means that those peaks only differ by their spin $\sigma$ and will combine in practice to yield modes with elliptical polarisation. In Fig. 4f, l, however, the separation between the rows of peaks is larger and this issue will not occur; each peak in the ($\omega/\sigma$, $\ell/\sigma$) spectrum then stands for a pure CP mode.

In summary, we can generate tuneable harmonic progressions with multiple levels of OAM per harmonic frequency, while both the frequency and the OAM levels of the harmonics can be controlled independently.

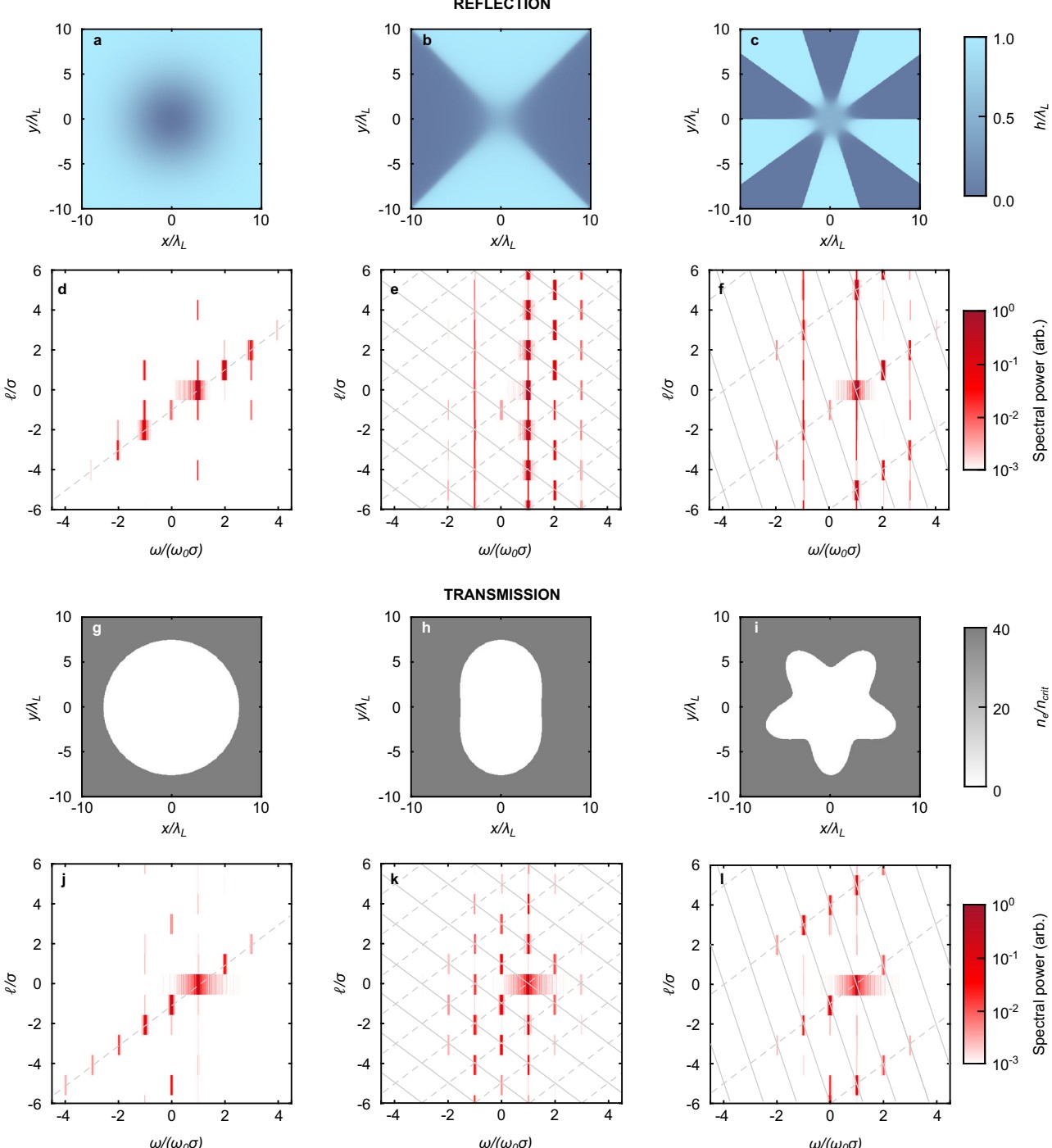

**Fig. 4 | Simulated harmonic spectra for laser beams interacting with structured targets. a–c** Surface height profiles of reflective targets with a conical dent ($q = 1$), and structures of $q = -1$ and $q = -4$, respectively. **d–f** The corresponding 2-D harmonic spectra from 3D simulations of these targets interacting with a CP pump pulse with $\ell = 0$. **g–l** Same, for a circular aperture and structured targets of 2- and 5-fold symmetry, respectively (equivalent to $q = 1$, $q = -1$ and $q = -4$), with the profiles displayed as cross-sectional density maps at $Z = 0.5\lambda_L$.

## The 2-D grid for $q \neq 1$: a tuneable frequency comb

In the above simulation results, targets with $q = 1$ are represented by $\ell/\sigma = -1$, while targets with $q \neq 1$ are represented by $\ell/\sigma = -1$ and $\ell/\sigma = -q$. In general, any target with $C_N$ symmetry (i.e. when the vector field $\nabla h$ or **n** is expressed in polar coordinates, its $\mathbf{e}_r$ and $\mathbf{e}_\varphi$ coefficients will be periodic in $\varphi$ with period $2\pi/N$) will usually lead to points $(0, -1)$ and $(0, N-1)$. Such a target corresponds to a function $\Psi_{DC} \propto \exp(-i\ell\varphi/\sigma)$ where $\Psi_{DC} \exp(-i\varphi)$ is $N$-periodic, so $\ell/\sigma + 1 = nN$, which includes the points $(0, -1)$ and $(0, N-1)$. We note that the structured target for a $q$-

plate has $C_{|q|+1}$ symmetry, see the Methods section for details. The same holds for an aperture target whose edge has $q$ lobes.

Any target with $C_q$ symmetry for $q \neq 1$ will thus be represented by two points in Fourier space. The pump laser adds at least a third point (when it is a pure CP mode) and together with the target will drive a 2-D grid in $(\omega/\sigma, \ell/\sigma)$ space, as shown above. When the rows of points are not parallel to the $\ell = 0$ axis, there will be a mode with $\ell = 0$ for only specific frequencies. This property can be used to obtain a frequency comb. In fact, any 2-D grid in $(\omega/\sigma, \ell/\sigma)$ space can potentially generate a

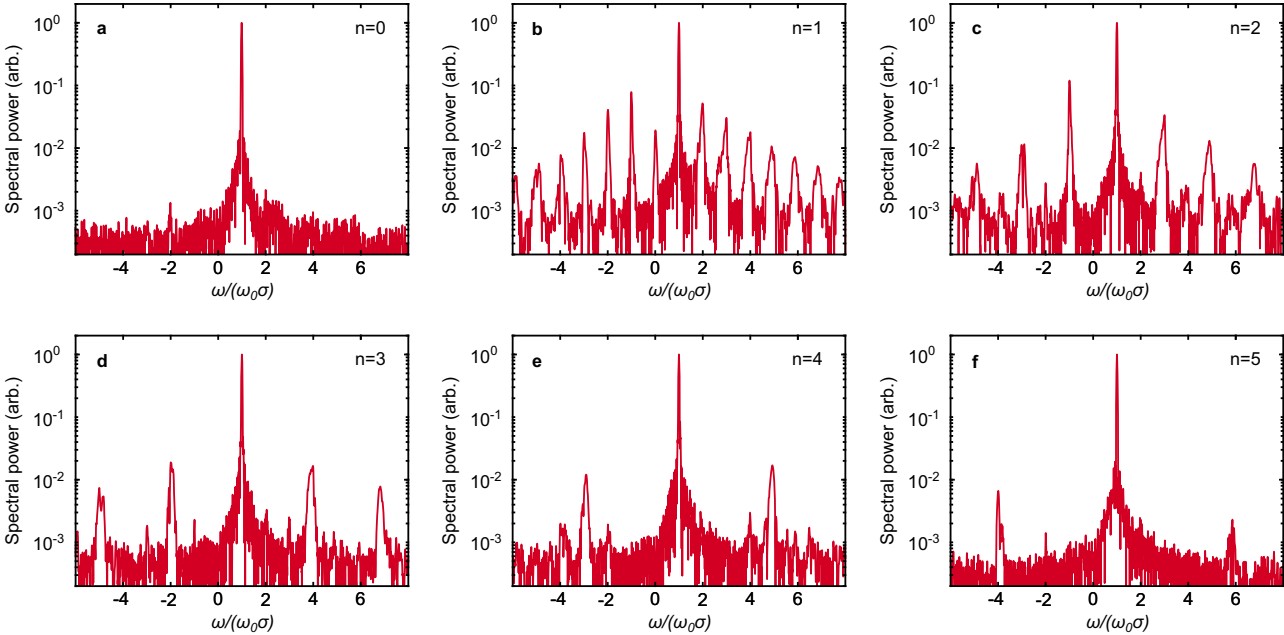

**Fig. 5 | Generation of frequency combs using aperture targets.** Frequency spectra generated at $y = 0$ and $x = 0$ from the interaction of a CP pump pulse with $\ell = 0$ with structured aperture targets with: **a**, $n = 0$ ($q = 1$); **b**, $n = 1$ ($q = 0$); **c**, $n = 2$ ($q = -1$); **d**, $n = 3$ ($q = -2$); **e**, $n = 4$ ($q = -3$); and, **f**, $n = 5$ ($q = -4$).

frequency comb (see, for example, Rego et al.[11]), and our methods can predict when this will happen. Also, a configuration showing a frequency comb usually contains a 2-D progression[11,45,46].

Any harmonic modes with $\ell = 0$ will have peak intensity on-axis, while harmonic modes with $\ell \neq 0$ will have an intensity minimum on-axis. By blocking out the off-axis HG, only those harmonic frequencies will be obtained that have an $\ell = 0$ component (and thus peak intensity on-axis). Such modes can be present in the harmonic spectrum even when all the driving modes have an intensity minimum on-axis[11]. For the case considered, for a nonlinear $q$-plate, the frequency step will be $|1 - q|\omega_0$, and the offset will be $\omega_0/\sigma_0$. The target shape can be independent of the exact laser wavelength and since we use plasma targets we can use this scheme for high-power laser systems, which have previously been shown to produce harmonics into the x-ray regime in laser-solid interactions[56].

In Fig. 5a–f, we show the Fourier spectra of the on-axis light (having $\ell = 0$) sampled behind aperture targets with $q = 1, 0, -1, -2, -3$ and $-4$ ($n = 0, ..., 5$), respectively. For $q = 1$ (circular aperture), we obtained a 1-D spectrum in Fig. 4j, and thus obtain only a single peak at the pump frequency. For other $q$, we obtain 2-D spectra (e.g. in Fig. 4k–l), and thus a frequency comb in $\omega/\sigma$ with spacing $\omega\sigma_0/(\sigma\omega_0) = |q - 1|$. For any $|q - 1| > 2$, the peaks do not occur symmetrically around 0, so all frequencies in the comb have circular polarisation. Adding OAM to the pump laser changes the shape of the 2-D harmonic grid and, thus, the frequencies of the comb.

## Discussion

Our approach enables investigation of harmonic generation in multiple variables in numerous laser-plasma interaction scenarios and beyond. These include: combining an OAM pump laser and a structured target to produce a plasma J-plate[57,58], tilting the target[15] or designing higher-order plasma $q$-plates[59] using magnetic multipoles. Our approach is also applicable to Raman scattering[28,60,61], beat wave generation[29,30] and more general HG scenarios in nonlinear optics[10,11,24,27]. Further details of these are provided in Supplemental section S6.

Generation of CP harmonics with tuneable OAM can also be achieved via two counter-rotating CP beams with frequencies

$\omega_2 = 2\omega_1 = 2(2\pi c/\lambda_L)$ ($\lambda_L = 800$ nm is used in most cases, corresponding to Ti:sapphire laser light), spin $\sigma_2 = -\sigma_1 = -1$ and OAM levels $\ell_1$ and $\ell_2$[10,24–26]. For this configuration, the harmonic peaks are regularly spaced along a single line in ($\omega/\sigma$, $\ell/\sigma$) space: $(\omega/(\sigma\omega_1), \ell/\sigma)_n = (1, \ell_1) + n(3, \ell_1 + \ell_2)$, which is tuneable via the choice of $\ell_{1,2}$. Configurations explored in previous work include $\ell_1 = 1$, $\ell_2 = 2$ (Xie et al.[25]), $\ell_1 = 1$, $\ell_2 = 1$ (Pisanty et al.[10]), $\ell_1 = 1$, $\ell_2 = -1$ and $\ell_1 = -2$, $\ell_2 = 1$ (Dorney et al.[24]), and $\ell_1 = 2$, $\ell_2 = 1$ (Minneker et al.[26]). Whilst all harmonic peaks are on a single line in some of these works, e.g., Pisanty et al.[10], the regular spacing observed in the present work is not achieved, nor is the variety of configurations. For the targets with a circular dent or aperture, the harmonic peaks are regularly spaced on a single line in ($\omega/\sigma$, $\ell/\sigma$) space, which can be tuned via the OAM level of the driving CP laser pulse. For the targets with one- to five-fold symmetry, the peaks form a 2-D regular grid, not seen in references[10,24–26], with the spacing between the diagonal rows equal to the symmetry level. Via our 2-D harmonic grid, we are also able to generate tuneable frequency combs (via choosing those harmonics with $\ell = 0$), something that cannot be achieved using the 1-D harmonic progressions used in references[10,24–26]. Our generic approach to harmonic generation can also be used to resolve apparent differences between the results reported in Fleischer et al.[36], which concludes a 1-D harmonic progression, and Pisanty et al.[42] and Milošević et al.[43], which suggest it is 3-D. Our framework shows that a harmonic progression generated by three independent CP modes will be a 2-D grid, since it has 2 degrees of freedom.

To conclude, we have developed a general approach to harmonic generation and harmonic progressions in ($\omega/\sigma$, $\mathbf{k}/\sigma$) space, in terms of the beating of "fundamental modes" with purely circular polarisation. We have demonstrated a method to analyse the harmonic ($\omega/\sigma$, $\mathbf{k}/\sigma$) spectrum via $\Psi = E_x + iE_y$ and the signed, multi-dimensional, complex Fourier transform of $\Psi$. We have reproduced earlier results on conical dent targets[12,13] and aperture targets[16], and extended these results to show, for the first time, the presence of "negative harmonics" with polarisation opposite to that of the pump. We have demonstrated the generation of a rich two-dimensional harmonic spectrum from a single CP laser interacting with structured reflective and aperture targets. Finally, we can now generate a tuneable harmonic frequency comb, via preferential selection of the harmonics ($\omega/\sigma$, 0) from the 2-D spectrum.

By revealing the underlying symmetry of these HG processes, this work can be readily applied to other laser-plasma interactions and to more general HG scenarios in nonlinear optics, simplifying analysis and facilitating a step-change in fundamental understanding.

## Methods

### Target description

We describe our reflective targets with a structured surface via a "height function" $h(x, y)$. We use $h(x, y) = 0$ for a flat target, $h(x,y) = x\tan\alpha$ for a tilted target, and $h(x,y) = \tan(\alpha)\sqrt{x^2 + y^2}$ for a target with a conical dent[12,13]. For a pump laser $\mathbf{A}_0$ hitting any non-flat target, the leading nonlinear factor (which drives the harmonics) is given by $\mathbf{A}_0 \cdot \nabla h$, where $\nabla h$ is constant in time but not in space. For a conical dent, we have $\nabla h \propto \mathbf{e}_r$ and obeys $\ell_{DC}/\sigma_{DC} = -1$. We aim to use structured surfaces whose height function contains modes with $\ell_{DC}/\sigma_{DC}$ different from 0 or $-1$.

For an aperture target, the leading nonlinear factor is given by $\mathbf{A}_0 \cdot \mathbf{n}$, where $\mathbf{n}$ is the normal vector of the inner surface of the aperture[16]. For a circular aperture, we have $\mathbf{n} = -\mathbf{e}_r$ and $\ell_{DC}/\sigma_{DC} = -1$. Different values of $\ell_{DC}/\sigma_{DC}$ can be obtained using apertures with different shapes.

For a pump laser $(\omega_0, \ell_0, \sigma_0)$ and a structured target with $\ell_{DC}/\sigma_{DC} = -q$, Eq. (1) yields: $\omega_n/\sigma_m = n\omega_0/\sigma_0$ and $\ell_n/\sigma_n + q = n$ $(\ell_0/\sigma_0 + q) = (\ell_0/\sigma_0 + q)(\omega_n/\sigma_m)/(\omega_0/\sigma_0)$. For $q = 1$, this expresses conservation of angular momentum within the EM waves. For $q \neq 1$, the HG process will leave angular momentum behind in the target.

For the "height function" of the reflective targets, we derive for $q \neq 1$ (details in the Supplemental section S7):

$$h(r,\varphi) = r^{1-q}\cos[(q-1)\varphi].$$

The case $q = 1$, covers both the conical shape of ref.[12] and the "spiral plate" of ref.[13]. For $q = 0$, we obtain $h(r,\varphi) = r\cos(\varphi) = x$, which corresponds to a laser beam with oblique incidence onto a plane, as described by Lichters et al.[33]. We see that we can design target surfaces to obtain the harmonic generation patterns corresponding to any $q$.

For the aperture targets, we use a "corrugated" inner surface $R(\varphi) = R_0 + dR\cos[(1-q)\varphi]$. This shape has been chosen such that the topology of $\mathbf{n}$ for a given value of $q$ matches the topology of $\nabla h$ for a "reflective" target with the same $q$.

### Simulation setup

Simulations were performed with EPOCH Particle-in-Cell code[62]. For the reflective targets, the simulation domain is defined as $-8\,\mu m \leq x, y, z \leq 8\,\mu m$ with $512 \times 512 \times 1024$ grid cells. The electron density is defined by the height function $z_{int} = h(x, y)$ such that $n(z < z_{int}) = 0$ and $n(z \geq z_{int}) = 4n_c$ where $n_c$ is the critical density calculated for a laser wavelength of $\lambda_L = 1\,\mu m$. In the reflective case the ions are immobile. The simulations are initialised with 5 particles per cell. The incoming laser pulse has a Gaussian profile with a half width of $w_0 = 4\,\mu m$ and a Gaussian temporal envelope with a duration of 100 fs. The wavelength is $\lambda = 1\,\mu m$ and the peak intensity is $I_{max} = 6.6 \times 10^{18}\,W\,cm^{-2}$. The boundaries are periodic in the $x$ and $y$-directions, and a perfectly matched layer absorbs the reflected laser pulse in the negative $z$-direction.

For the aperture targets, the simulation domain is defined as $-10\,\mu m \leq x, y \leq 10\,\mu m$ and $-5\,\mu m \leq z \leq 15\,\mu m$ with $720 \times 720 \times 1000$ grid cells. All boundaries were defined as free space. The plasma was composed of a slab of $Al^{13+}$ ions of thickness $\lambda_L$ with an empty circular aperture with a periodically varying radius, $R(\varphi)$ with $R_0 = 3.75\lambda_L$ and $dR = 0.75\lambda_L$. This was neutralised with an electron population with a peak density equal to $40n_c$ and an initial temperature of 10 keV. The front surface of the target was set at $z = 0$. Each species was initialised with 80 particles per cell. The laser pulse was circularly polarised ($\sigma_0 = 1$, $l_0 = 0$) in the $[x, y]$ direction and focused with a Gaussian spatial profile at $z = 0$

resulting in $w_0 = 4\,\mu m$. The temporal profile was also defined as Gaussian with a FWHM of 40 fs. The peak intensity was $I_{max} = 1 \times 10^{19}\,W\,cm^{-2}$. The slightly higher laser intensity than the reflective target case was choosen because higher signal-to-noise ratio is acheived and therefore, the harmonic features are more visible. This is discussed in Supplemental Section S8, which includes simulation results for the transmission case that have been run with the same parameters as the reflective case ($I_{max} = 6.6 \times 10^{18}\,W\,cm^{-2}$ and 100 fs) to enable direct comparison.

## Data availability

The data that support the figures within this paper and other findings of this study are available at https://doi.org/10.15129/09b8b425-b71e-4f07-ad7a-b163946670e9.

## Code availability

The EPOCH particle-in-cell code used in this work is available at https://epochpic.github.io/.

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

## Acknowledgements

This work was financially supported by EPSRC, Grant nos. EP/R006202/1 (P.McK. and M.K.) and EP/V049232/1 (P.Mc.K). It involved the use of the ARCHER2 high-performance computer with access provided via the Plasma Physics HEC Consortia, Grant no. EP/X035336/1 (P.McK.). EPOCH was developed under EPSRC Grant no. EP/G054940/1.

## Author contributions

R.T. developed the basic concepts, analytic theory, target design, and data analysis. H.S and M.K. performed the simulations and implemented the data analysis. R.B. and P.McK. contributed to the development of the concepts and understanding of the underlying physics. All authors contributed to the preparation of the manuscript.

## Competing interests

The authors declare no competing interests.
