## [Peer Review File · Nature Communications]

REVIEWERS' COMMENTS

Reviewer #1 (Remarks to the Author):

The authors have placed considerable effort into improving the quality of the presented manuscript during the revision process. Specifically, they have addressed the comments of both reviewers by including expanded analysis, additional simulations, and including additional text in the main text and SI. The additional far-field simulation results (Figure 3 in main text) add a lot of confidence to the proposed generation scheme for producing unique SAM and OAM states as it is clear that the harmonics would survive into the far-field. The authors also take time to address the possibility of separating and analyzing harmonics experimentally, which, even if uncommon, provides additional framework for this work to be adopted by the experimental community. The authors also take more time to explain their findings and frame them in the landscape of existing literature (as shown by the additional references included in the revised text), which helps to further showcase the novelty of their approach. I believe with these additions, as well as the extended discussion to clarify the points raised by the reviewers, will make this work accessible to a larger audience such as that of Nature Communications. The novelty is also well explained and, in my opinion, meets the level of the the journal. For these reasons, I recommend to publish the article as is.

Reviewer #2 (Remarks to the Author):

In their revised manuscript and and replies the authors have robustly addressed my concerns in relation to the oversight of Xie et al. and a clearer physical picture for how the corrugated surfaces can impact the efficiency but not the spectral content of the emitted beam that were not clear in their original submission. These two points are core to enhancing the impact of the present work on the broad readership of Nature Communications.

As a result I am satisfied that the revised submission meets the the criteria for publication in Nature Communications.